# Impact of Sublethal Insecticides Exposure on *Vespa magnifica*: Insights from Physiological and Transcriptomic Analyses

**DOI:** 10.3390/insects15110839

**Published:** 2024-10-25

**Authors:** Qingmei Hu, Sijia Fan, Kaiqing Liu, Feng Shi, Xueting Cao, Yiquan Lin, Renyuan Meng, Zichao Liu

**Affiliations:** School of Agriculture and Life Sciences, Kunming University, Kunming 650214, China; hwxhqm476@163.com (Q.H.); 19513337601@163.com (S.F.); lkqkm@163.com (K.L.); sf14787788954@163.com (F.S.); 18768702317@163.com (X.C.); lin523231314999@sina.com (Y.L.); 17854335708@163.com (R.M.)

**Keywords:** *Vespa magnifica*, sublethal concentration, developmental calendar, transcriptome

## Abstract

Insecticides, while boosting crop yields, can harm non-target insects like *Vespa magnifica*. This study explored the effects of four common insecticides: thiamethoxam, avermectin, chlorfenapyr, and β-cypermethrin, on *Vespa magnifica*. Although larval survival was unchanged, pupation and fledge rates dropped significantly. Enzyme assays showed stress responses with increased antioxidant activity and suppressed peroxidase levels. Transcriptomic analysis revealed heightened energy expenditure, impacting essential functions like flight and immunity. Key pathways related to metabolism and nervous system activity were also affected, potentially impairing learning, memory, and detoxification. These results highlight the broader ecological risks of insecticide exposure, emphasizing the need for better strategies to protect beneficial insect populations.

## 1. Introduction

Pesticides play a crucial role in modern agriculture and are widely used worldwide to control pests and protect crop yields. These chemicals include insecticides, acaricides, nematicides, herbicides, and plant growth regulators [1]. While effective against pests, insecticides also pose significant risks to non-target beneficial insects [2]. In the midst of the Sixth Great Species Extinction, the decline in arthropod populations underscores the urgency of understanding insecticides impacts on these species [3]. For instance, imidacloprid exposure lowers activity levels in *Bombus impatiens*, and phoxim and cypermethrin trigger detoxification enzymes in *Meteorus pulchricornis* [4,5]. However, the effects of insecticides on *Vespa magnifica* remain largely unexplored.

*Vespa magnifica*, a species of the order Hymenoptera, suborder Apocrita, family Vespidae and genus *Vespa*, is a holometabolous insect that undergoes four developmental stages: egg, larva, pupa, and adult. Both females and drones die before winter with only the queen surviving to overwinter and re-emerge in early March. The queen then constructs a nest and begins laying eggs. The full developmental cycle spans 36 to 43 d, consisting of 6–8 d for the egg stage, 10–15 d for the larval stage, and 16–20 d for the pupal stage [6]. The developmental timeline of *Vespa magnifica* is significantly influenced by factors such as the generation number, nutrient intake, and local climate [7]. As ground-nesting wasps, *Vespa magnifica* are exposed to insecticides throughout their life cycle, including during foraging, nesting, nursing, and hibernation. They can also be exposed to insecticides present in the air, plants, propolis, resins, and soil [8]. Exposure to insecticides disrupts the nervous system of *Vespa magnifica*, causing disorientation and lethargy, which in turn reduces hunting and nest-building efficiency. Additionally, it impairs development, particularly pupation and eclosion, leading to delays and abnormalities in larvae, thus decreasing their ability to successfully pupate and emerge as adults [9,10].

*Vespa magnifica* plays significant ecological and economic roles [11]. Ecologically, as a predatory insect, it helps regulate populations of other insects, including pests that can damage crops or disrupt ecosystems. For instance, in 1978, Shimen County in Hunan Province, China, successfully utilized wasps to control cotton bollworm infestations, dramatically reducing insecticide costs and usage [12]. Similarly, in the Shizuoka Prefecture, Japan, wasps were used for pest management in tea plantations, while in Feng County, Jiangsu Province, wasps were employed to control cotton bollworms, achieving a pest elimination rate of 98.5% [13]. Building on the homing behavior of *Vespa velutina*, Guo et al. designed portable wooden or wire mesh boxes for rearing hornets. These boxes, housing hornet colonies, were strategically placed in vegetable, fruit, and tea fields, enabling fast, thorough, and environmentally safe pest control [14]. Additionally, *Vespa magnifica* can prey on various pests such as *seudaulacaspis pentagona*, *Cetonia aurata*, *Plutella xylostella*, *Locusta migratoria*, *Propylaea japonica*, and *Orius strigicollis* [6]. Economically, *Vespa magnifica* indirectly benefits agriculture by reducing the need for chemical insecticides, leading to cost savings and more sustainable farming practices [15]. Additionally, its presence in ecosystems can enhance pollination indirectly by controlling herbivorous insects, contributing to healthier crop yields [16].

Thiamethoxam, a second-generation neonicotinoid insecticide, is considered the most widely used insecticide globally due to its high efficiency, broad-spectrum activity, and strong systemic properties. It acts on the nicotinic acetylcholine receptors in the insect nervous system. The 25% water-dispersible granule formulation of thiamethoxam is commonly used against pests such as *Aonidiella aurantii* (Hemiptera), *Sitobion avenae* (Aphididae), and *Popillia japonica* (Coleoptera) [17,18]. In recent years, the negative effects of thiamethoxam on non-target organisms, particularly bees, have garnered significant attention. For instance, prolonged or short-term exposure to thiamethoxam has been shown to impair bee flight ability [19]. Additionally, thiamethoxam causes sublethal effects in bees, damaging their central nervous system and leading to midgut injuries [20]. In social wasps, exposure to lethal and sublethal concentrations of thiamethoxam reduces mobility and disrupts the stability of worker wasps and the colony [21]. Brazilian wasps (*Polybia paulista*) exposed to LC_50_ and LC_10_ sublethal concentrations of thiamethoxam exhibit reduced activity, loss of the intestinal brush border, the presence of spherocrystals, and brain cell apoptosis characterized by loss of cell contact and pyknosis [22]. In response to these concerns, the European Union fully banned the registration of thiamethoxam for use outside permanent greenhouses in 2018 and proposed the revocation of all maximum residue limits for thiamethoxam on agricultural products in 2022 [23]. As a hymenopteran of economic significance, studying the toxicity of thiamethoxam on *Vespa mandarinia* could provide a solid research foundation and robust scientific data to support adjustments to China’s neonicotinoid insecticide registration policies and future technical trade negotiations with the EU and other members regarding neonicotinoid-related measures.

Avermectin, a macrolide antibiotic produced by streptomyces avermitilis, is known for its unique mechanism of action and strong insecticidal activity, exhibiting both stomach toxicity and contact effects. With the increasing use of 1.8% avermectin emulsifiable concentrate, concerns have arisen regarding its potential toxicity to non-target organisms in the environment [24]. This highlights the need to evaluate the sublethal effects of avermectin on wasps, particularly *Vespa magnifica*. Understanding the sublethal effects of avermectin is crucial, as these effects may not result in immediate mortality but can significantly impact the health and behavior of wasp populations. For instance, studies have shown that exposure to sublethal doses of avermectin can lead to alterations in metabolic functions in other species, such as reduced glycogen levels and lipid synthesis, along with the inhibition of key enzyme activities [25]. These metabolic disruptions can compromise energy balance, reproductive success, and overall fitness. In addition, avermectin has been found to impair cell proliferation and induce mitochondrial apoptosis in honeybee cells [26]. Such findings underscore the potential risk that avermectin poses to wasps, which may experience similar cellular and physiological disturbances. The induction of toxic effects and the disruption of normal physiological functions could lead to decreased foraging efficiency, impaired nest maintenance, and weakened colony dynamics. By elucidating the mechanisms through which avermectin affects wasps, we can better assess its risks to beneficial insect populations and contribute to the development of more sustainable pest management practices.

Chlorfenapyr is a novel aryl-substituted pyrrole compound with a broad spectrum of insecticidal activity, offering safety and long-lasting effects by targeting insect mitochondria [27]. The 12% emamectin benzoate–chlorfenapyr suspension is highly effective in controlling Orthopteran pests, such as locusts and katydids, as well as Lepidopteran pests, which are key prey for *Vespa magnifica* [28]. Field studies on sublethal toxicity in *Osmia excavata* exposed to chlorfenapyr showed significant delays in larval development and a marked reduction in emergence rates [29]. While the sublethal effects of chlorfenapyr on *Apis mellifera* have been documented [30], research on its effects on *Vespa magnifica* remains limited. Thus, evaluating the sublethal effects of chlorfenapyr on *Vespa magnifica* is crucial for assessing its ecological risks especially given the critical predatory and ecological role of *Vespa magnifica*.

β-cypermethrin is a widely used pyrethroid insecticide that primarily exerts its insecticidal effect by disrupting the sodium ion channels in the nervous system, leading to impaired movement, spasms, paralysis, and eventually death [31]. However, sublethal doses of β-cypermethrin pose negative impacts on non-target insects, particularly important predatory species like *Vespa magnifica*. Studies have shown that sublethal doses of β-cypermethrin not only contaminate honeybee hives, causing severe harm to the colonies, but also inhibit the growth and development of other insects, such as the significant weight reduction observed in *Spodoptera litura* larvae [32,33]. Additionally, in natural ecosystems, pests like *Solenopsis invicta* are known to threaten bee colonies during breeding seasons, and the application of insecticides may interfere with *Vespa magnifica*’s predatory behavior and population stability [34]. Therefore, assessing the sublethal effects of β-cypermethrin on *Vespa magnifica* is crucial for understanding its ecological risks and providing a scientific foundation for insecticide management policies.

The aim of this study was to evaluate the effects of sublethal concentrations (LC_10_, LC_20_, and LC_30_) of thiamethoxam, avermectin, chlorfenapyr, and β-cypermethrin on the physiological development of *Vespa magnifica* using a feeding exposure method and to further investigate these impacts through transcriptome analysis. Utilizing a feeding exposure method, we assessed parameters such as larval survival, pupation rates, fledge success, body weight at various developmental stages, and enzyme activity. Advanced transcriptome analysis was conducted to pinpoint differentially expressed genes (DEGs), which was followed by in-depth gene ontology (GO) and Kyoto Encyclopedia of Genes and Genomes (KEGG) pathway analyses, with results further validated through reverse transcription quantitative polymerase chain reaction (RT-qPCR). This comprehensive approach provides a valuable foundation for further ecological risk assessments and the development of targeted pest management that mitigates unintended impacts on non-target species such as *Vespa magnifica*.

## 2. Materials and Methods

### 2.1. Ethical Guidelines

*Vespa magnifica* is neither a protected nor endangered species and is commonly found in rural and suburban areas of southern China. According to the legislation of the People’s Republic of China, research involving arthropods does not require ethical approval [35]. Our housing conditions and testing procedures comply with established animal welfare standards.

### 2.2. Animals

The *Vespa magnifica* specimens were acquired from the Wasp General Base at the Yunnan (Lufeng) Hatchery Park (102.016751° N, 25.043597° E) and reared at the School of Agriculture and Life Sciences, Kunming University. Rearing conditions were kept at a temperature of (25 ± 2) °C and a relative humidity of (60 ± 10)%. The diet included a nutritional solution (Wasp General Base at the Yunnan (Lufeng) Hatchery Park, Chuxiong, China), honey, and *Locusta migratoria manilensis* (Maoyuan Locust Breeding Base, Kunming, China). The experiment, conducted during the egg-laying period of the queen, used healthy one-day-old larvae. All specimens originated from the same colony and were starved for 2 h before the experiment.

### 2.3. Insecticides

In this study, commercial formulations were used instead of technical drug insecticides to reflect real-world exposure in *Vespa magnifica* habitats, providing more accurate data for ecological risk assessment and management [36]. According to our preliminary survey results, the insecticides used in this study are commonly applied in *Vespa magnifica* breeding areas across nine cities in Yunnan, China: Kunming, Dali, Chuxiong, Yuxi, Qujing, Mengzi, Pu’er, Xishuangbanna, and Mangshi [37]. These insecticides include thiamethoxam (25% water-dispersible granules, Chuan Dong Pesticide & Chemical Co., Ltd., Dazhou, China), avermectin (1.8% emulsifiable concentrate, Shandong Rongbang Chemical Co., Ltd., Binzhou, China), chlorfenapyr (12% suspension concentrate, Shandong Yilan Technology Co., Ltd., Weifang, China), and β-cypermethrin (4.5% microemulsion, Chengdu Bangnong Chemical Co., Ltd., Chengdu, China). Based on our previous acute toxicity studies, the LC_10_, LC_20_, and LC_30_ sublethal concentrations of the insecticides thiamethoxam, avermectin, chlorfenapyr, and β-cypermethrin were determined (Table 1).

### 2.4. Sublethal Effects of Thiamethoxam, Avermectin, Chlorfenapyr, and β-Cypermethrin on the Developmental Stages of Vespa magnifica

A basic larval diet was prepared consisting of 50% nutritional solution, 37% sterile water, 6% glucose, 6% locust juice, and 1% yeast extract, which can be stored at 4 °C for 3 days [38]. The locust juice was prepared by removing the wings and legs of locusts, which was followed by crushing them in a blender. The test insecticides formulations were dissolved in ultrapure water to create stock solutions to replace sterile water in the preparation of the larval diet (Table 1). Insecticide exposure was conducted within the combs of 1-day-old *Vespa magnifica* larvae. The larval combs were randomly divided into five groups with each group marked and consisting of 50 larvae: (1) control group (larval diet with sterile water); (2) thiamethoxam group (larval diet containing LC_10_, LC_20_, and LC_30_ concentrations of thiamethoxam); (3) avermectin group (larval diet containing LC_10_, LC_20_, and LC_30_ concentrations of avermectin); (4) chlorfenapyr group (larval diet containing LC_10_, LC_20_, and LC_30_ concentrations of chlorfenapyr); (5) β-cypermethrin group (larval diet containing LC_10_, LC_20_, and LC_30_ concentrations of β-cypermethrin). Every morning at 9:00 a.m., the combs were removed, and 20 µL of different insecticide solutions was fed to the larvae using a dropper. The combs were then returned to the *Vespa magnifica* breeding box with adult workers. Feeding continued for six days until the larvae were capped. During this period, adult *Vespa magnifica* were given a 50% honey solution and Locusta migratoria manilensis, and in turn, they fed the larvae. Observations were made at 12 h intervals (12 h, 24 h, 36 h, 48 h, 54 h, 60 h, 66 h, and 72 h). The number of dead larvae, pupae, and newly fledged *Vespa magnifica* was recorded with dead individuals promptly removed to prevent pathogen growth. Larval survival, pupation, and fledge rates were calculated, and body weights were measured. The artificial climate chamber was controlled by an air conditioner (KFR-35GW, Midea, China) and a humidifier (CS-3VWL, Midea, China) with the experiment conducted in an artificial climate chamber at 25 ± 2 °C and 50–60% relative humidity.

### 2.5. Evaluation of Antioxidant Enzyme and Peroxidase Activity in Vespa magnifica Exposed to Thiamethoxam, Avermectin, Chlorfenapyr, and β-Cypermethrin

To prepare 10 mL of LC_10_, LC_20_, and LC_30_ insecticide solutions for thiamethoxam, avermectin, chlorfenapyr, and β-cypermethrin, equal parts honey and water (1:1) were mixed. The control group received the same honey–water mixture. Each treatment was replicated three times with 20 wasps per group. *Vespa magnifica* were first fed the insecticide solution, which was followed by continuous access to the honey-water mixture. After 24 h, surviving wasps were stored at −80 °C for further analysis.

To prepare wasp abdominal worm specimens, 0.5 g of tissue was weighed, frozen in liquid nitrogen, and ground. After grinding, 5 mL of the corresponding enzyme kit extraction buffer was added. The mixture was then homogenized in an ice bath after thawing. It was centrifuged at 8000× *g* for 10 min at 4 °C using a high-speed refrigerated centrifuge (TGL-1650, Shuke Instrument Co., Ltd., Chengdu, China). The supernatant was collected, with one portion kept on ice for immediate analysis, and the remainder was frozen at −80 °C for future use. Superoxide dismutase (SOD) activity was measured at 560 nm, catalase (CAT) activity at 240 nm, and peroxidase (POD) activity at 470 nm using enzyme assay kits (Solarbio Science & Technology Co., Ltd., Beijing, China) on a spectrophotometer (CARY60, Shimadzu International Trading Co., Ltd., Shanghai, China).

### 2.6. RNA Sequencing

The impact on the *Vespa magnifica* became more pronounced as the sublethal concentrations increased [37]. To investigate the potential molecular mechanisms underlying *Vespa magnifica* exposure to thiamethoxam, avermectin, chlorfenapyr, and β-cypermethrin, we constructed a total of 15 sequencing libraries from adult *Vespa magnifica* treated with these insecticides. Specifically, *Vespa magnifica* were exposed to sublethal concentrations (LC_10_) of thiamethoxam, abamectin, chlorfenapyr, and β-cypermethrin for 24 h, while the control group received 50% honey water. After treatment, the surviving hornets were immediately dissected to extract the intestines, which were collected into 1.5 mL centrifuge tubes, treated with liquid nitrogen, and stored at −80 °C. Total RNA was extracted using the TaKaRa column kit (Baori Medical Biotechnology, Beijing, China), and RNA concentration and purity were measured with a Nanodrop^TM^ (Nanodrop 2000, Thermo Fisher Scientific, Waltham, MA, USA). The RNA Integrity Number (RIN) was determined using the Agilent 2100 (G2938C, Agilent, Santa Clara, CA, USA). For single library construction, the total RNA must meet the following requirements: ≥1.0 µg in total amount, concentration ≥ 35.0 ng/µL, and OD 260/280 ≥ 1.8. A library was constructed using more than 1.0 µg of total RNA, which included mRNA purification, mRNA fragmentation, cDNA synthesis, end repair and A-tailing, adapter ligation, library selection, and quality control [39]. RNA sequencing and library construction were performed at Shanghai Meiji Biomedical Technology Co., Ltd., Shanghai, China.

### 2.7. Transcriptome Sequencing, Gene Annotation and DEG Analysis

The raw data were obtained using the sequencing by synthesis (SBS) technology. Illumina HiSeq 2000 was used to analyze the data, removing adapter-containing reads, reads with undetermined base information, and low-quality reads. The effective reads (clean reads) for further analysis were obtained by checking the GC content and base error rate. Quality control of the raw data was performed using a Fastx_toolkit (http://hannonlab.cshl.edu/fastx_toolkit/, 0.0.14, accessed on 15 October 2023). Hisat 2 (http://ccb.jhu.edu/software/hisat2/index.shtml, 2.1.0, accessed on 20 October 2023) was used as the alignment tool for sequence alignment to the reference genome [40]. Gene annotation was performed using Swiss-Prot (ftp://ftp.uniprot.org/pub/databases/uniprot/current_release/knowledgebase/complete/uniprot_sprot.fasta.gz, 2022.10, accessed on 20 October 2023). Kallisto (https://pachterlab.github.io/kallisto/download, 0.46.0, accessed on 25 October 2023) was employed to quantify gene expression levels [41]. DESeq2 (http://bioconductor.org/packages/stats/bioc/DESeq2/, 1.24.0, accessed on 25 October 2023) was used to analyze the differential expression between the control group and the groups treated with thiamethoxam, avermectin, chlorfenapyr, and β-cypermethrin [42]. GO enrichment and KEGG pathway analyses for DEGs were conducted using Goatools (https://files.pythonhosted.org/packages/bb/7b/0c76e3511a79879606672e0741095a891dfb98cd63b1530ed8c51d406cda/goatools-0.8.9.tar.gz, 0.6.5, accessed on 30 October 2023).

### 2.8. Reverse Transcription Quantitative Polymerase Chain Reaction (RT-qPCR)

In insects, cytochrome P450 (CYP450) plays a crucial role in detoxification, breaking down harmful chemicals and reducing their toxicity. Selecting CYP450-related genes for RT-qPCR validation in *Vespa magnifica* provides insight into the activation of detoxification pathways in response to insecticide exposure, making these genes key markers for understanding the molecular mechanisms of insecticide resistance and tolerance [43]. GAPDH, a key enzyme in glycolysis, plays a vital role in cellular metabolism and is minimally influenced by environmental factors. Therefore, selecting GAPDH as the optimal internal reference gene for gene expression studies in the gut of *Vespa magnifica* ensures both the accuracy and consistency of experimental results [44]. To further verify the accuracy of the gene expression trends observed in RNA sequencing, 9 genes related to the CYP450 enzyme system were randomly selected from the DEGs for qRT-PCR validation (Table 2). The RT-qPCR reactions were performed using the SYBR^®^ Premix Ex Taq™ II (Tli RNaseH Plus) kit (TaKaRa, Ōtsu City, Japan) and a real-time quantitative PCR instrument (StepOne^TM^ 4376373, ThermoFisher, Waltham, MA, USA). The reaction system consisted of 10 μL of 2X ChamQ SYBR Color qPCR Master Mix, 0.8 μL of each forward and reverse primer, 10 μL of 50X ROX Reference Dye, 6 μL of ddH_2_O, and 2 μL of diluted cDNA. The RT-qPCR conditions were 95 °C for 5 min, followed by 40 cycles of 95 °C for 30 s, 55 °C for 30 s, and 72 °C for 40 s [45]. The relative expression of target genes was calculated using the comparative CT method (Relative expression = 2^−ΔΔCT^) [46].

### 2.9. Statistical Analysis

SPSS v27.0 (SPSS Inc., Chicago, IL, USA) software was used to perform a Shapiro–Wilk test for normality and Levene test for chi-square. Significant analysis of variance (*p* < 0.05) was carried out using one-way ANOVA for variance and Tukey’s method after passing the test. The significance of differences in the larval survival rate, pupation rate, fledge rate, body weight and enzyme activity were calculated. Origin v21.0 drawing software was used for drawing.

## 3. Results

### 3.1. Effect of Sublethal Effects of Insecticides Exposure on Development, Body Weight, and Enzyme Activity in Vespa magnifica

Under sublethal stress, thiamethoxam had a significant effect on the survival rate of *Vespa magnifica* larvae, while abamectin, chlorfenapyr, and β-cypermethrin did not show significant differences in larval survival rates. However, thiamethoxam, abamectin, chlorfenapyr, and β-cypermethrin all had significant effects on pupation and fledge rates with the differences becoming more pronounced as sublethal concentrations increased. Among them, the thiamethoxam LC_30_ treatment group showed the most notable reductions with pupation and eclosion rates significantly decreasing by 36.99% and 52.06%, respectively (*p* < 0.05, Figure 1A).

During the larval stage, larvae treated with thiamethoxam at LC_30_ showed a 25.9% increase in body weight compared to the control group (2.28 ± 0.018 g). In contrast, larvae treated with avermectin, chlorfenapyr, and β-cypermethrin exhibited lower body weights than the control, although their weights increased with higher concentrations. Across all treatments, body weights remained consistently lower than controls during both the pupal and fledging stages. No significant differences in body weight were observed between the chlorfenapyr and β-cypermethrin groups at emergence, which was possibly due to environmental adaptability in the insects (*p* < 0.05, Figure 1B).

As the sublethal concentrations increased, the activity levels of SOD and CAT in *Vespa magnifica* treated with thiamethoxam, abamectin, and chlorfenapyr for 24 h initially rose and then declined compared to the control group. At the sublethal concentration of LC_20_, SOD activity peaked at 1396.21 U/g, and CAT activity reached a maximum of 986.26 U/g. In contrast, after 24 h of exposure to β-cypermethrin, SOD and CAT activity initially decreased and then increased with SOD activity peaking at 1106.20 U/g and CAT activity at 1137.81 U/g. Both enzymes exhibited activation effects. Additionally, as sublethal concentrations increased, the activity of POD in *Vespa magnifica* exposed to thiamethoxam, abamectin, chlorfenapyr, and β-cypermethrin for 24 h gradually decreased compared to the control group, showing an inhibitory effect on POD activity. At the sublethal concentration of LC_30_, thiamethoxam caused the highest inhibition of POD activity, reaching 425.60 U/g (*p* < 0.05, Figure 1C).

### 3.2. Transcriptome Sequencing, Gene Annotation and Analysis of DEGs

The total RNA yield for each sample was ≥1.0 µg with concentrations ≥35.0 ng/µL, an OD 260/280 ratio of ≥1.8, and an OD 260/230 ratio of ≥1.0. Raw reads from *Vespa magnifica* intestinal transcriptome sequencing ranged from 41.18 to 50.87 million. After quality control, 97.8% of the raw reads were retained (40.86 to 50.37 million) with Q20 and Q30 values exceeding 95% and 93%, respectively (Table 3). All genes and transcripts from the *Vespa magnifica* transcriptome were compared against six databases (NR, Swiss-Prot, Pfam, EggNOG, GO, KEGG) using BLAST. NR annotations showed the highest proportion (58.23%) under sublethal insecticide stress with 40.92% of sequences aligning with *Vespa mandarinia* (Figure 2A,B).

Volcano plot analysis revealed 3628 DEGs between the thiamethoxam and control groups, with 3572 upregulated and 55 downregulated (Figure 3A), indicating a strong upregulatory effect. Avermectin exposure resulted in 359 DEGs (350 upregulated, 9 downregulated) (Figure 3B), showing a milder impact compared to thiamethoxam. Chlorfenapyr led to 14 DEGs (4 up-regulated, 10 downregulated) (Figure 3C), primarily causing downregulation. β-cypermethrin had minimal influence, with 10 DEGs (one upregulated, nine downregulated) (Figure 3D). Cluster analysis grouped DEGs with similar expression patterns, which was likely associated with related metabolic and signaling pathways (Figure 3E).

GO enrichment analysis of the DEGs revealed significant enrichment in biological process (BP), cellular component (CC), and molecular function (MF) categories, providing a comprehensive and detailed insight into the roles of genes within the organism [47]. In the BP category, the DEGs between thiamethoxam, abamectin, bromo-fipronil, β-cypermethrin, and the control group were primarily enriched in the subcategories of cellular process and metabolic process. In the CC category, the DEGs were mainly enriched in cell part, protein-containing complex, and organelles. In the MF category, DEGs were predominantly enriched in binding and catalytic activity (Figure 4A–D).

In the transcriptome sequencing of the gut of *Vespa magnifica,* the DEGs were potentially involved in various functions such as detoxification, metabolism, and immunity. KEGG pathway enrichment analysis further revealed how these genes participate in specific metabolic pathways, signal transduction processes, or biological networks [48]. Compared to the control, thiamethoxam treatment resulted in the enrichment of 1371 KEGG metabolic pathways, involving 695 DEGs. The most significantly enriched pathways included the cell cycle, mRNA surveillance pathway, pentose phosphate pathway, nucleotide excision repair, basal transcription factors, HIF-1 signaling pathway, AMPK signaling pathway, and ribosome biogenesis in eukaryotes (Figure 4E). For avermectin, 462 KEGG pathways were enriched compared to the control group, involving 367 DEGs. The top ten enriched pathways possibly linked to detoxification properties included ribosome, glycolysis/gluconeogenesis, the pentose phosphate pathway, starch and sucrose metabolism, and pyruvate metabolism (Figure 4F). Chlorfenapyr treatment led to the enrichment of eight KEGG pathways, involving eight DEGs. The significantly enriched pathways included cholinergic synapse, pantothenate and CoA biosynthesis, beta-alanine metabolism, glycerophospholipid metabolism, and nucleocytoplasmic transport (Figure 4G). For β-cypermethrin, nine KEGG pathways were enriched, involving 21 DEGs. The significantly enriched pathways included herpes simplex virus 1 infection, cholinergic synapse, antigen processing and presentation, retrograde endocannabinoid signaling, and glycerophospholipid metabolism (Figure 4H).

The RT-qPCR results were consistent with the RNA-seq data, confirming the reliability of the transcriptome analysis (Figure 5).

## 4. Discussion

In this study, thiamethoxam, avermectin, chlorfenapyr, and β-cypermethrin had no significant effect on the survival of *Vespa magnifica* larvae. However, there were significant variations in pupation and fledge rates, which increased at sublethal concentrations. Notably, exposure to sublethal concentrations of insecticides, particularly thiamethoxam at LC_30_, led to a reduction in pupation rates to 36.99% and fledge rates to 52.06%, which can be attributed to various physiological disruptions. Thiamethoxam interacts with nicotinic acetylcholine receptors, causing neurotoxicity and developmental disorders [49]. In the LC_30_ treatment group, the reduction in pupation and fledge rates indicates a severe impairment in the larvae’s ability to undergo metamorphosis. Additionally, the energy required for detoxification diverts resources from critical developmental processes, further contributing to the decrease in both pupation and fledge rates [50]. Other insecticides, such as avermectin, chlorfenapyr, and β-cypermethrin, although less toxic to larval survival, still significantly affected pupation and fledge rates. As sublethal concentrations increased, the cumulative effects on pupation and fledge became more pronounced, reflecting the broader impact of sublethal stress on the development of *Vespa magnifica*. These findings align with research on *Bombus terrestris* larvae exposed to sublethal dinotefuran and reduced pupation rates in crabronid wasps exposed to thiamethoxam and imidacloprid [51,52]. Similar effects were observed in honey bees treated with chlorantraniliprole and propiconazole [53]. These findings emphasize the need to consider sublethal insecticide exposure when evaluating ecological risks, particularly for beneficial species like *Vespa magnifica*. In *Vespa magnifica* larvae exposed to LC_30_ concentrations of thiamethoxam, weight gain was observed. Thiamethoxam disrupts the endocrine system in larvae, affecting growth regulation and increasing the energy demands for detoxification, which leads to higher nutrient intake or storage [54]. After exposure, *Vespa magnifica* larvae alter their feeding behavior, redirecting energy toward detoxification and resulting in weight gain.

Sublethal insecticide exposure, particularly thiamethoxam, abamectin, chlorfenapyr, and β-cypermethrin, significantly impacted the enzyme activities of SOD, CAT, and POD in *Vespa magnifica,* which are critical for managing oxidative stress [55]. SOD and CAT, which neutralize harmful reactive oxygen species (ROS) [56], initially increased in response to thiamethoxam, abamectin, and chlorfenapyr, peaking at LC20 (SOD at 1396.21 U/g and CAT at 986.26 U/g). This suggests an early defense against oxidative stress. However, enzyme activity declined at higher concentrations, which was likely due to enzyme saturation or damage from prolonged stress [57]. β-cypermethrin showed a delayed response with SOD and CAT activity first decreasing and then peaking at lower concentrations (SOD at 1106.20 U/g and CAT at 1137.81 U/g), indicating a different oxidative stress pattern. POD, essential for breaking down peroxides, consistently decreased across all insecticide treatments, showing the strongest inhibition with thiamethoxam at LC30 (425.60 U/g). Since POD is responsible for breaking down peroxides, its inhibition implies that the *Vespa magnifica* are less capable of detoxifying these harmful by-products. As a result, the sustained high levels of ROS could lead to oxidative damage, impairing critical biological processes in adult *Vespa magnifica*, such as mobility, foraging, and overall survival [58]. Similar trends were observed in *Microplitis mediator*, *Helicoverpa armigera*, *Diadegma semiclausum*, and *Spodoptera exigua* when exposed to various insecticides [59,60]. Overall, these findings demonstrate that while SOD and CAT are initially upregulated to counteract insecticide-induced oxidative stress, prolonged exposure overwhelms these defenses. The inhibited POD activity further exacerbates oxidative damage, contributing to impaired functions in adult *Vespa magnifica*, such as reduced mobility, foraging ability, and overall survival.

The use of intestinal tissue for RNA sequencing in *Vespa magnifica* is essential because the gut plays a critical role in physiological processes like digestion, immunity, and detoxification. It is also the primary site for toxin exposure and response when studying the effects of insecticides [61]. By analyzing RNA expression in the gut, we can uncover molecular mechanisms involved in detoxification, immune responses, and other metabolic pathways affected by sublethal insecticide exposure [62]. This approach has been successfully applied in *Apis mellifera* studies to reveal how environmental stressors influence gene expression patterns in the gut [63]. This study employed high-throughput sequencing to analyze intestinal gene expression in *Vespa magnifica* under four insecticide stressors, revealing molecular responses and transcriptional regulation.

Thiamethoxam caused the strongest transcriptional response, with 3628 DEGs, mostly upregulated, affecting pathways like cell cycle, mRNA surveillance, and AMPK signaling, which are key to energy regulation [64]. This suggests increased energy demand and oxidative stress under thiamethoxam exposure. Research has shown similar transcriptional responses in other insects exposed to thiamethoxam, especially regarding gene expression changes linked to energy metabolism and oxidative stress. For example, a study on *Apis mellifera* exposed to thiamethoxam demonstrated significant changes in gene expression, particularly in genes associated with detoxification and stress responses [65]. Pathways such as energy production and oxidative stress management were notably affected, supporting findings that insecticide exposure can disrupt cellular processes essential for survival and adaptation.

Avermectin had a milder effect on *Vespa magnifica*, causing significant changes with 359 DEGs, but the numbers of upregulated genes and affected pathways were fewer than with thiamethoxam. The enriched pathways, including the ribosome, glycolysis, pentose phosphate pathway, and pyruvate metabolism, are crucial for energy generation and carbohydrate metabolism [66]. This suggests that under avermectin exposure, *Vespa magnifica* may adjust its metabolism to enhance energy utilization and carbohydrate breakdown in response to increased energy demands. Similar metabolic adjustments have been reported in other insects. For example, studies on *Helicoverpa armigera* revealed an altered expression in energy production pathways, including glycolysis and pyruvate metabolism, highlighting a metabolic shift to meet insecticide-induced energy demands [67]. Research on *Spodoptera frugiperda* also demonstrated significant changes in gene expression related to glycolysis and ATP synthesis following avermectin exposure, reinforcing the findings in *Vespa magnifica* [68].

The chlorfenapyr treatment altered the expression of 14 DEGs in *Vespa magnifica*, with 10 genes downregulated, indicating an inhibitory effect. KEGG analysis revealed few affected pathways, including cholinergic synapse and β-alanine metabolism. The cholinergic synapse enrichment suggests potential disruptions in neural signaling that could indirectly impact energy metabolism, while β-alanine metabolism indicates a possible inhibition of energy acquisition processes [69]. Similar studies have documented significant transcriptional changes in other insects due to chlorfenapyr. For example, research on *Bombyx mori* found a downregulation of genes related to metabolic processes and energy regulation [70]. Another study on *Frankliniella occidentalis* reported alterations in gene expression affecting amino acid metabolism [71]. This indicates that the insecticide can inhibit processes crucial for energy acquisition, affecting the insect’s overall energy balance.

β-cypermethrin had a minimal impact on *Vespa magnifica*, affecting only 10 DEGs, which were primarily downregulated. KEGG enrichment analysis revealed pathways such as herpes simplex virus 1 infection and antigen presentation, indicating a focus on immune responses rather than direct energy metabolism. Similar effects have been observed in other insects. For instance, a study on *Spodoptera litura* showed that exposure to β-cypermethrin mainly influenced immune-related genes with little impact on metabolic pathways [72]. Additionally, research on *Solenopsis invicta* also indicated alterations in gene expression related to immune response mechanisms and stress pathways [73], further suggesting a minimal effect on energy metabolism compared to other insecticides.

Across all insecticides, DEGs were enriched in functions related to detoxification, metabolism, and immunity. ATP-related processes, essential for maintaining energy balance, were consistently upregulated in response to all treatments, indicating an increased energy demand for detoxification. This included the upregulation of purine metabolism, nucleoside triphosphate catabolism, and mitochondrial electron transport. Overall, *Vespa magnifica* exhibits varying degrees of resistance and adaptation to these insecticides through targeted gene regulation. Notably, the mitochondrial electron transport in *Vespa mandarinia* is significantly enhanced when exposed to insecticides, which is crucial for ATP synthesis and physiological functions [74]. Insufficient energy can impair flight, hunting, and immunity, reducing the organism’s ability to detoxify insecticides and increasing mortality risk.

Detoxification metabolism enables insects such as *Vespa magnifica* to handle external toxins by converting lipophilic substances into water-soluble forms for easier excretion [75]. The identification of nine DEGs related to the CYP450 enzyme system in *Vespa magnifica* under various LC10 insecticide treatments highlights the critical role of detoxification pathways in the organism’s response to chemical stressors. These genes, including members of the CYP4, CYP4V, CYP6, CYP61, and CYP9 families, as well as GST, PPIB, and UTG, are known for their involvement in detoxification and xenobiotic metabolism [76]. The observed upregulation of these genes suggests that *Vespa magnifica* relies heavily on the CYP450 system and associated detoxification mechanisms to mitigate the effects of insecticide exposure.

## 5. Conclusions

This study highlights the sublethal effects of various insecticides, including thiamethoxam, abamectin, chlorfenapyr, and β-cypermethrin, on the development, body weight, enzyme activity, and gene expression in *Vespa magnifica*. While larval survival rates showed no significant changes, both pupation and fledge rates significantly declined as insecticide concentrations increased. Additionally, the body weight of *Vespa magnifica* increased during the larval, pupal, and fledge stages. The activation of SOD and CAT enzymes, along with the inhibition of POD in adult *Vespa magnifica*, indicates the triggering of oxidative stress.

At the molecular level, thiamethoxam induced the strongest transcriptional response with 3628 DEGs significantly upregulated. Pathways related to ATP processes and mitochondrial electron transport were notably affected, reflecting the increased energy demands for detoxification. Avermectin and chlorfenapyr triggered milder transcriptional responses with key metabolic pathways such as glycolysis, pentose phosphate, and pyruvate metabolism being upregulated to meet the energy needs of the *Vespa magnifica*. In contrast, β-cypermethrin had the least impact with key pathways like herpes simplex virus 1 infection and antigen presentation being downregulated, affecting immune responses. The regulation of detoxification pathways, particularly the CYP450 enzyme system, highlights *Vespa magnifica*’s response to insecticide stress.

In conclusion, this study uncovers the sublethal effects of various insecticides on development, body weight, enzyme activity, and gene expression in *Vespa magnifica* with significant differences observed in detoxification processes and energy metabolism regulation. These findings emphasize the broader ecological risks of insecticide exposure to non-target insects and underscore the need for further research on the long-term effects of newer insecticides and strategies to protect beneficial insect populations.

## Figures and Tables

**Figure 1 insects-15-00839-f001:**
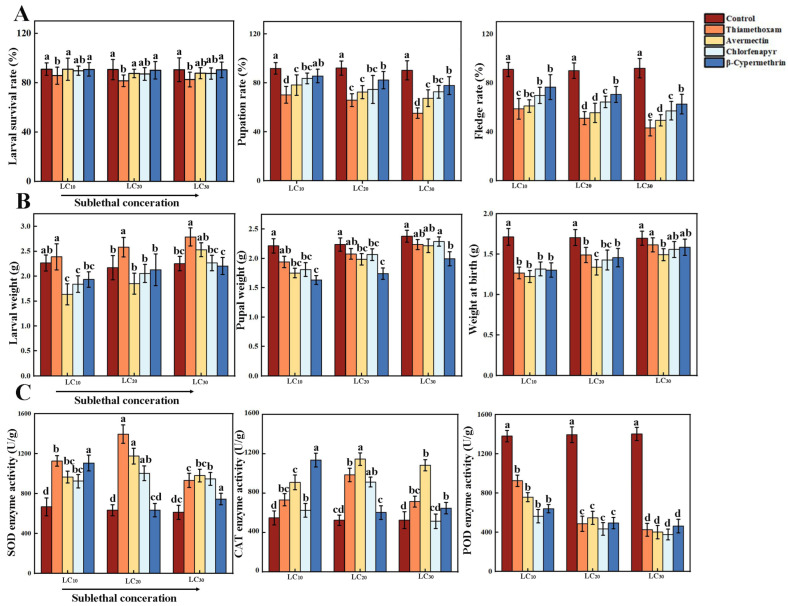
Sublethal effects of insecticides exposure on development, body weight, and enzyme activity in *Vespa magnifica*. (**A**) Sublethal effects of insecticides exposure on larval survival, pupation and fledge rates of *Vespa magnifica* (*n* = 50, *p* < 0.05). (**B**) Body weight changes in *Vespa magnifica* larvae treated with sublethal concentrations of insecticides during the larval, pupal, and fledge stages (*n* = 50, *p* < 0.05). (**C**) SOD, CAT, and POD enzyme activity in *Vespa magnifica* following exposure to sublethal concentrations of insecticides (*n* = 20, *p* < 0.05). The different letters (a, b, c and d) above the bars indicate significant differences between the groups.

**Figure 2 insects-15-00839-f002:**
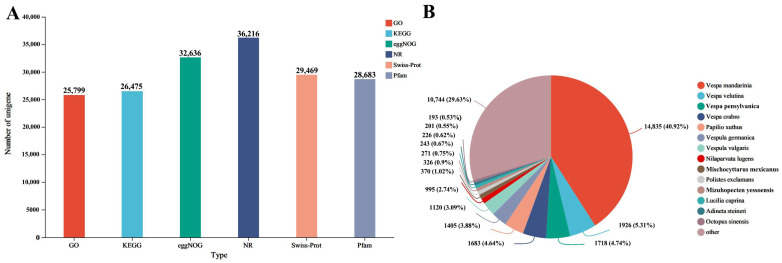
Functional annotation and species distribution of unigenes in *Vespa magnifica* transcriptome. (**A**) Annotation result statistics of unigenes of *Vespa magnifica* transcriptome in different databases. (**B**) Species distribution of unigenes of *Vespa magnifica* NR database.

**Figure 3 insects-15-00839-f003:**
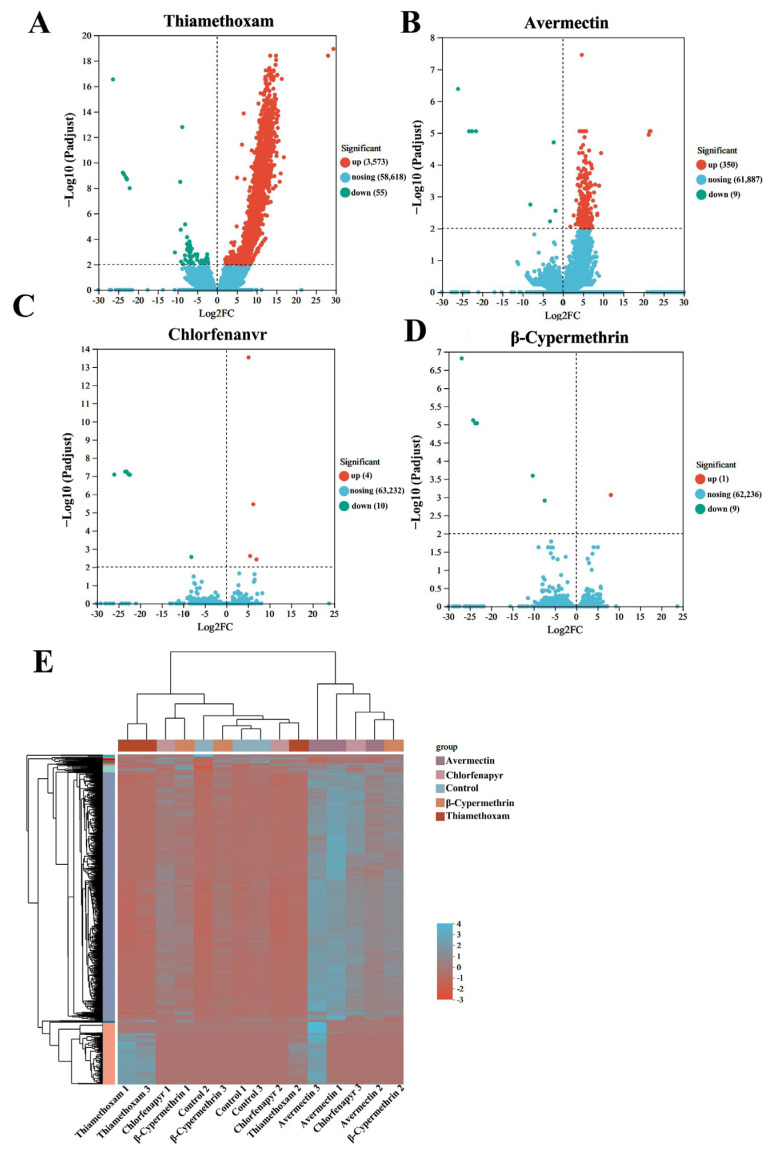
DEGs between the control and four insecticides exposure for 24 h. (**A**) DEGs induced by thiamethoxam at an LC_10_ dose. (**B**) DEGs induced by avermectin at an LC_10_ dose. (**C**) DEGs induced by chlorfenapyr at an LC_10_ dose. (**D**) DEGs induced by β-cypermethrin at an LC_10_ dose. (**E**) Cluster analysis of DEGs of *Vespa magnifica*.

**Figure 4 insects-15-00839-f004:**
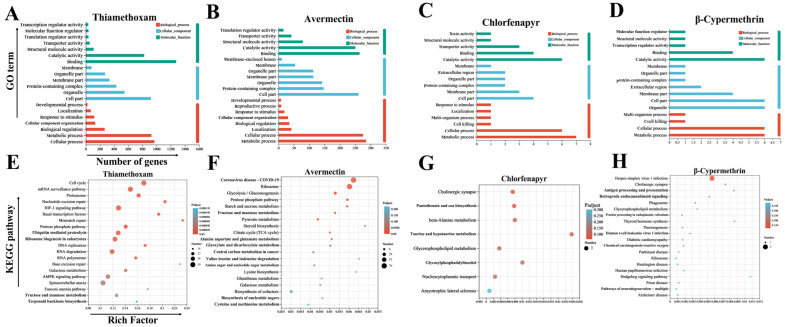
GO and KEGG enrichment analysis of DEGs reveals key pathways and functions affected by insecticide treatments in *Vespa magnifica*. (**A**) GO aggregation map of DEGs in thiamethoxam and control. (**B**) GO aggregation map of DEGs in avermectin and control. (**C**) GO aggregation map of DEGs in chlorfenapyr and control. (**D**) GO aggregation map of DEGs in β-cypermethrin and control. (**E**) Enrichment analysis of KEGG pathway of thiamethoxam and control DEGs. (**F**) Enrichment analysis of KEGG pathway of avermectin and control DEGs. (**G**) Enrichment analysis of KEGG pathway of chlorfenapyr and control DEGs. (**H**) KEGG pathway enrichment analysis of cypermethrin and control DEGs.

**Figure 5 insects-15-00839-f005:**
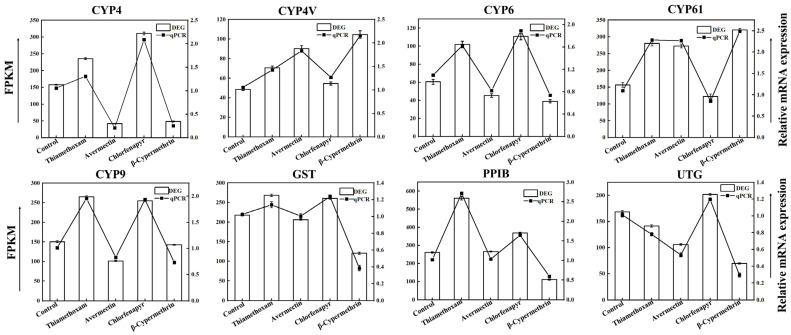
qPCR validation of nine DEGs related to the CYP450 enzyme system confirms consistency with RNA-seq data.

**Table 1 insects-15-00839-t001:** Different sublethal concentrations of four insecticides to *Vespa magnifica* [37].

Insecticides	Sublethal Concentrations (μg a.i./wasp)		
	LC_10_	LC_20_	LC_30_
Thiamethoxam	2.42 × 10^−6^	5.07 × 10^−6^	6.34 × 10^−6^
β-Cypermethrin	8.24 × 10^−5^	1.74 × 10−^4^	3.09 × 10^−4^
Avermectin	2.28 × 10^−6^	4.42 × 10^−6^	6.29 × 10^−6^
Chlorfenapyr	6.43 × 10^−6^	1.06 × 10^−5^	1.80 × 10^−5^

**Table 2 insects-15-00839-t002:** Sequences of primers used for qRT-PCR analysis.

Gene ID	KO Name	NCBI	Primer F	Primer R	Length (bp)
NODE_cov_102.801070_g8279_i0	CYP4	XP_035741425.1	TTGTCCAGCCATATTTAC	TAGGTGCTATTAGTTTACGA	127
NODE_cov_98.805527_g8081_i0	CYP4V	XP_035742762.1	CGAAGCCATTCATAAACA	ATACTCCACGGTCTCCTC	125
NODE_cov_68.538621_g2989_i0	CYP6	XP_035741324.1	TGAATGTATGGTTCCCAGTT	CAATCCGAAGGGCAAGTA	137
NODE_cov_74.509154_g2989_i0	CYP61	CAF1632358.1	CTATACGAATTGGCTCTG	GAAATGTTACTGGTGGAT	169
NODE_cov_61.292950_g3449_i0	CYP9	XP_035722235.1	CTTATTGCGTACTTGTCC	AATGATACCCTTCTCGTC	127
NODE_cov_53.046424_g4522_i0	GST	XP_035723860.1	GCTAACAACAGGACCATC	AGCCCATAATACAATAACCA	149
NODE_cov_80.862000_g15569_i0	PPIB	KAH9408598.1	TCCAAATTGGCGGTAAAG	TTGATAACCATCCAGCAC	270
NODE_cov_74.683208_g15569_i0	UTG	XP_046836138.1	ACATAGACCCATTATCACC	TACCCATAAGACCACCAT	300
NODE_cov_16.023701_g14314_i0	GAPDH	KAI2807735.1	CGATGTTCGTCGTTGGTG	TTTGGGTTGCCGTGATAG	171

**Table 3 insects-15-00839-t003:** Transcriptome sequencing data.

Sample	Raw Reads	Clean Reads	Average Base Error Rate (%)	Q20 (%)	Q30 (%)	GC (%)
Control 1	45,917,960	45,562,310	0.0273	96.27	94.11	35.97
Control 2	44,923,142	44,359,422	0.0272	96.25	94.16	39.05
Control 3	41,176,086	40,861,948	0.0276	95.92	93.66	37.23
Thiamethoxam 1	46,030,496	45,668,930	0.0278	96.26	94.07	36.54
Thiamethoxam 2	45,640,090	45,245,878	0.0269	96.32	94.18	34.08
Thiamethoxam 3	46,618,708	46,124,634	0.0272	96.19	93.91	30.88
Avermectin 1	50,873,080	50,371,152	0.0274	96.06	93.85	36.79
Avermectin 2	42,207,426	41,867,502	0.0278	95.85	93.55	36.72
Avermectin 3	48,891,090	48,240,970	0.0269	96.35	94.08	32.15
Chlorfenapyr 1	41,813,518	41,388,414	0.0279	95.82	93.46	36.91
Chlorfenapyr 2	46,337,574	45,928,132	0.0279	95.82	93.49	37.68
Chlorfenapyr 3	46,115,384	45,661,982	0.0273	96.17	93.88	36.75
β-Cypermethrin 1	44,129,178	43,744,382	0.0275	96.23	93.76	38.20
β-Cypermethrin 2	44,483,326	44,100,214	0.0274	96.05	93.81	37.62
β-Cypermethrin 3	46,039,540	45,601,858	0.0273	96.09	93.89	36.16

## Data Availability

All the sample raw reads obtained in this study have been deposited at the National Center for Biotechnology Information (accession number: PRJNA1141747).

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
