# Peer review of "Impact of Sublethal Insecticides Exposure on Vespa magnifica: Insights from Physiological and Transcriptomic Analyses"

_insects, 2024, doi:10.3390/insects15110839_

Round 1

Reviewer 1 Report

Comments and Suggestions for Authors

The authors of the article under discussion present important information about issues related to the health of Vespa magnifica when exposed to insecticides. To make the text clearer and more coherent, I present the following suggestions:

1. Regarding the title: since four active ingredients used as insecticides were utilized, the term 'pesticide' should be changed to 'insecticide', making the title more precise and faithful to the content of the paper.

2. Regarding the simple summary: what POD means? This should be described since it is appearing in the text for the first time. The initial approach of this section, as well as the abstract, introduction, and even the discussion, are similar. It needs to be changed to avoid repetition.

3. Regarding the Introduction: A report is made on the effect of a fungicide on another species of wasp. Since the present study only addresses insecticides, the approach should consider only insecticides (lines 43 and 44).

The taxonomic presentation of the studied species should follow a sequence from the more general to the more specific (lines 48 and 49).

To highlight the importance of the studied species, it is necessary to describe its ecological and economic functions.

There is no connection between the topics addressed in the paragraphs. They need to be linked, following a logical sequence. There are short and fragmented sentences.

4. Regarding Material and Methods: 
The description of the insect's life cycle is not appropriate for this section. It can be relocated to the Introduction or Discussion. In any case, it should be followed by a reference.

There is no description of the evaluations related to the life cycle of the insects (item 2.2).

What does "nutrient solution" means (lines 120-121). 

Subdivide the item 2.3 into Animals and Insecticides.

The title for 2.4 item must be rewritten.

Pharmacological treatment? Does it make sense?

Make it clear: was the exposure to insecticides done once or daily through food? (lines 140-141).

The insecticides doses must be cited.

Item 2.5 Determination of biological activity... : incomplete title

Was the food consumption measured? Why not?

It is necessary to improve the description of the larval maintenance method.

Why was the intestine the tissue used for RNA sequencing? Comment in the discussion. Use a reference to support your choice.

Also for item 2.6, does it only address LC10 and the other two doses?

Table 2 should present references for each gene ID.

5. Regarding the Results:

"Although the larval survival rates of Vespa magnifica exposed to sublethal concentrations of thiamethoxam, avermectin, chlorfenapyr, and β-cypermethrin were comparable". 

It is always comparable. The point here is whether it is statistically different or not.

There are results for variables not described in the materials and methods. Please review.

6. Regarding the Discussion:

The beginning is repetitive; the first paragraph could be excluded. The authors should place greater emphasis on the most important results. Line 329: In this study... Overall, this item needs improvement. There are too many results for a superficial approach.

7. Regarding the Conclusion:
it is neither a new summary nor a continuation of the discussion. The interpretation of the results should be presented here in a direct, clear, and objective manner.

Comments on the Quality of English Language

Minor editing of English language required.

Author Response

Dear reviewer,

Thank you for reviewing our manuscript and for the valuable comments, which greatly helped us to improve the manuscript. The manuscript was carefully revised and point-by-point response was listed below. We hope that your comments have been addressed accurately. The revised manuscript was marked with yellow color and the responses were presented in blue text.

Comments 1: Regarding the title: since four active ingredients used as insecticides were utilized, the term 'pesticide' should be changed to 'insecticide', making the title more precise and faithful to the content of the paper.

Response 1: We have changed 'pesticide' to 'insecticide', and the full text has been revised.

Comments 2: Regarding the simple summary: what POD means? This should be described since it is appearing in the text for the first time. The initial approach of this section, as well as the abstract, introduction, and even the discussion, are similar. It needs to be changed to avoid repetition.

Response 2: POD refers to peroxidase. We have described since it is appearing in the text for the first time.

Comments 3:

(1)Regarding the Introduction: A report is made on the effect of a fungicide on another species of wasp. Since the present study only addresses insecticides, the approach should consider only insecticides (lines 43 and 44).

(2)The taxonomic presentation of the studied species should follow a sequence from the more general to the more specific (lines 48 and 49).

(3) To highlight the importance of the studied species, it is necessary to describe its ecological and economic functions.

(4)There is no connection between the topics addressed in the paragraphs. They need to be linked, following a logical sequence. There are short and fragmented sentences.

Response 3:

(1) Regarding the Introduction: A report is made on the effect of a fungicide on another species of wasp. Since the present study only addresses insecticides, the approach should consider only insecticides (lines 43 and 44).

In the introduction of this study, only pesticides were considered and fungicides were removed at the teacher's prompt. (lines 44-46)

(2) The taxonomic presentation of the studied species should follow a sequence from the more general to the more specific (lines 48 and 49).

We have studied the order in which the taxonomic representation of species should follow from more general to more specific (lines 48 and 49).

(3) To highlight the importance of the studied species, it is necessary to describe its ecological and economic functions.

We have highlight the importance of the studied species and describe its ecological and economic functions (lines 63-78).

(4) There is no connection between the topics addressed in the paragraphs. They need to be linked, following a logical sequence. There are short and fragmented sentences.

We have restated the paragraphs in a logical order.

Comments 4:

(1) Regarding Material and Methods: The description of the insect's life cycle is not appropriate for this section. It can be relocated to the Introduction or Discussion. In any case, it should be followed by a reference.

(2) There is no description of the evaluations related to the life cycle of the insects (item 2.2).

(3) What does "nutrient solution" means (lines 120-121).

(4) Subdivide the item 2.3 into Animals and Insecticides.

(5) The title for 2.4 item must be rewritten.

(6) Pharmacological treatment? Does it make sense?

(7) Make it clear: was the exposure to insecticides done once or daily through food? (lines 140-141).

(8) The insecticides doses must be cited.

(9) Item 2.5 Determination of biological activity... : incomplete title

(10) Was the food consumption measured? Why not?

(11) It is necessary to improve the description of the larval maintenance method.

(12) Why was the intestine the tissue used for RNA sequencing? Comment in the discussion.

(13) Use a reference to support your choice.

(14) Also for item 2.6, does it only address LC10 and the other two doses?

(15) Table 2 should present references for each gene ID.

Response 4:

(1) Regarding Material and Methods: The description of the insect's life cycle is not appropriate for this section. It can be relocated to the Introduction or Discussion. In any case, it should be followed by a reference.

Regarding materials and methods: we have relocated the description of the Vespa magnifica life cycle to the Introduction and added references.

(2) There is no description of the evaluations related to the life cycle of the insects (item 2.2).

It is a description of assessments related to life cycles (item 2.3).

(3) What does "nutrient solution" means (lines 120-121).

"Nutritional solution" refers to a mixture of specific vitamins and honey used to formulate larval feed. It is purchased from the Yunnan Lufeng Wasp Hatchery Base and is developed and supervised by the Beijing Institute of Entomology. The source has been added to the article (lines 169-170).

(4) Subdivide the item 2.3 into Animals and Insecticides.

We have subdivided item 2.3 into animals and pesticides.

(5) The title for 2.4 item must be rewritten.

The title for 2.4 item have be rewritten (lines 190-191)

(6) Pharmacological treatment? Does it make sense?

Pharmacological treatment is not suitable, we have modify it in the text (lines 197-198).

(7) Make it clear: was the exposure to insecticides done once or daily through food? (lines 140-141).

We have make it clear: was the exposure to insecticides done daily through food (lines 204-207).

(8) The insecticides doses must be cited.

The insecticides doses have be cited (line 189).

(9) Item 2.5 Determination of biological activity: incomplete title

Item 2.5 has been rewritten to make it more complete (lines 215-216).

(10) Was the food consumption measured? Why not?

During the experiment, the feeding of the marked larvae was carried out by worker wasps of Vespa magnifica, making it impossible to determine the exact amount of food consumed by the larvae. The worker wasps, being housed in a hive and present in large numbers, also had their individual food intake unmeasured. Therefore, the food consumption was not measured.

(11) It is necessary to improve the description of the larval maintenance method.

We have improve the description of the laeval maintenance method (lines 192-209).

(12) Why was the intestine the tissue used for RNA sequencing? Comment in the discussion.

We have commented in the discussion about using intestinal tissue for RNA sequencing (lines 460-467).

(13) Use a reference to support your choice.

We have use a reference to support our choice (line 194).

(14) Also for item 2.6, does it only address LC10 and the other two doses?

Also for item 2.6, As the sublethal concentrations of the four insecticides increased, their effects on Vespa magnifica became more pronounced. In this study, we focused on the impact of the lowest dose, using LC10, as described and cited in the text (lines 236-242).

(15) Table 2 should present references for each gene ID.

We have provided references for each gene ID in Table 2.

Comments 5: Regarding the Results:

(1) "Although the larval survival rates of Vespa magnifica exposed to sublethal concentrations of thiamethoxam, avermectin, chlorfenapyr, and β-cypermethrin were comparable".

(2) It is always comparable. The point here is whether it is statistically different or not.

Response 5:

(1) "Although the larval survival rates of Vespa magnifica exposed to sublethal concentrations of thiamethoxam, avermectin, chlorfenapyr, and β-cypermethrin were comparable".

"Although the larval survival rates of Vespa magnifica exposed to sublethal concentrations of thiamethoxam, avermectin, chlorfenapyr, and β-cypermethrin were comparable". Only thiamethoxam had a significant effect, while abamectin, chlorfenapyr, and cypermethrin showed no significant impact, as revised in the text (lines 303-309).

(2) There are results for variables not described in the materials and methods. Please review.

We have reviewed the results and confirmed their consistency with the materials and methods (materials and methods lines 212-214; results lines 303-317).

Comments 6: Regarding the Discussion:

The beginning is repetitive; the first paragraph could be excluded. The authors should place greater emphasis on the most important results. Line 329: In this study... Overall, this item needs improvement. There are too many results for a superficial approach.

Response 6: Regarding the discussion: The beginning has been excluded.  The discussion has been rewritten with improvements (lines 460-535).

Comments 7: Regarding the Conclusion:

it is neither a new summary nor a continuation of the discussion. The interpretation of the results should be presented here in a direct, clear, and objective manner.

Response 7: We have rewritten the conclusion to make it more direct and clear (lines 537-560).

Reviewer 2 Report

Comments and Suggestions for Authors

The manuscript " Impact of sublethal pesticide exposure on Vespa magnifica: insights from physiological and transcriptomic analyses". The topic is interesting enough to investigate and this research provides important insights into poorly understood sublethal effects of pesticides on the health of non-target insects. There were, however, a few issues with the statistical approach and the presentation of some results as described below. Additionally, some discussion points need further elaboration.

1. Line 2, the “Vespa magnifica” should be italics.

2. Line 99, the conservation strategies.

3. Line 107, the paragraph “2.2 Vespa magnifica life cycle” should be placed in the introduction.

4. Line 113-114, the references about the influence of pesticides on Vespa magnific should be supplied and also a short introdcution should be introduced.

5. Line 125-127, is this the result of your survey or is it supported by relevant literature?

6. Line 127-132, the pesticides used in this research were commercial formulations, why not use technical drugs?

7. Line 132-133 the grammar of this sentence “sublethal concentrations for each of the four pesticides were determined” should be modified.

8. Line 140, the daily dose was 2 μL which was not enough for larvae of different ages.

9. Line 145-146, the final concentration of the solution was LC20/2?

10. Line 148, what the dose of the pesticide solution.

11. Line 152, milliliters? or mL?

12. In Materials and Methods, please supply the detail information of the Instruments and equipment, such as manufacturer and equipment model etc.

13. Line 162, what is the number of intestinal tracts of wasps used for research ? Why not a whole wasp, but just choose the intestines.

14. Line 166-167, as this work being done by you or biological company, if it is a company, please supply the name of the company.

15. Line 177, what is the reason for choosing CYP450 gene? The full name of CYP450 should be provide.

16. Line 181, the 2 in ddH2O should be subscripted.

17. Line 183, why GAPDH was selected as the internal reference gene? However, the Real time quantitative PCR (RT-qPCR) methods are inadequate. The authors should refer to Bustin et al. 2009 (The MIQE Guidelines: Minimum Information for Publication of Quantitative Real-Time PCR Experiments. Clinical Chemistry 55:4611-622). In general, at least two or more internal reference genes must be provided to achieve accurate normalization, because the transcript levels of reference genes are not always stable and vary with the developmental stage and organization of the insect as well as the experimental conditions.

18. Line 184, supply the reference of “CT method”.

19. Line 189 and 193, supply the information about SPSS and Origin.

20. Line 195, 220, 221, Sublethal effects of insecticides

21. Line 203, why does thiamethoxam LC30 cause weight gain in larvae ? and also a further discussion.

22. Line 211-213, this sentence is difficult to read.

23. Line 220, due to the different mechanisms of action of the different insecticides, the horizontal coordinates are classified according to the pesticides in Figure 1, while the results of the different treatments are expressed in coloured bars, which makes it easier for the reader to know the differences between treatments of the same active ingredient at different concentrations.

24. Line 228-229, this sentence is not clear, is it the RNA of all the samples or just one of them?

25. Line 268-272, this section is duplicated in the Materials and Methods section, and it is sufficient to state only the results, which are suggested for deletion.

26. Line 302-304, it is possible to analyse the causes of the decrease in pupation rates in the different treatment groups.

27. Line 306, the object of reference 31 and 32 is not Apis mellifera mellifera, please check.

28. Line 310-313, it is appropriate to talk about the effects of insecticides on the three enzymes in terms of their functions.

29. Line 316, can ROS changes be caused by four insecticides? add the appropriate literature description.

30. Line 325, this citation [38] is inappropriate and more literature needs to be reviewed and the effects of pharmaceuticals on their energy metabolism discussed.

31. Line 329-339, In-depth analysis of transcriptome results for each insecticide in relation to its mechanism of action.

32. Line 334, is there a statistics on food consumption in this study?

33. Line 342, Is there any relevant literature on Vespa magnifica has fewer detoxification genes compared to other species. If so, please add.

34. I don't really agree with the conclusion that the alteration  of these genes (CYP4, CYP4V, CYP6, CYP61, CYP9, PPIB, GST, and UTG) leads to immune disorders and impact the overall physiological state of Vespa magnific.

35. The format of the references does not meet the requirements of the journal and needs to be modified. Such as volume and issue need to be in italics, a comma between the page number and the volume and issue. Please check them one by one.

Comments on the Quality of English Language

Careful editing of the grammar and sentence structure to enhance readability. Some issues concerning lack of clarity are highlighted in the reviewer comments but more extensive editing is recommended.

Author Response

Dear reviewer,

Thank you for reviewing our manuscript and for the valuable comments, which greatly helped us to improve the manuscript. The manuscript was carefully revised and point-by-point response was listed below. We hope that your comments have been addressed accurately. The revised manuscript was marked with yellow color and the responses were presented in blue text.

Comments 1. Line 2, the“Vespa magnifica”should be italics.

Response 1: the“Vespa magnifica”has been italics (Line 2).

Comments 2. Line 99, the conservation strategies.

Response 2: the conservation strategies has been rewritten (Lines 155-157).

Comments 3. Line 107, the paragraph “2.2 Vespa magnifica life cycle” should be placed in the introduction.

Response 3: the paragraph “2.2 Vespa magnifica life cycle” have been placed in the introduction (Lines 48-57).

Comments 4. Line 113-114, the references about the influence of pesticides on Vespa magnific should be supplied and also a short introdcution should be introduced.

Response 4: We have added references regarding the influence of pesticides on Vespa magnifica and included a brief introduction (lines 55-61).

Comments 5. Line 125-127, is this the result of your survey or is it supported by relevant literature?

Response 5: This is our surey of result, we have rewritten the sentence and added references to previous work (lines 178-181).

Comments 6. Line 127-132, the pesticides used in this research were commercial formulations, why not use technical drugs?

Response 6: In this study, commercial formulations were used instead of technical drug insecticides to reflect real-world exposure in Vespa magnifica habitats, providing more accurate data for ecological risk assessment and management. We have mentioned in the text and attached references (lines 176-178).

Comments 7. Line 132-133 the grammar of this sentence “sublethal concentrations for each of the four pesticides were determined” should be modified.

Response 7: We have modified the grammar of this sentence “sublethal concentrations for each of the four pesticides were determined” (lines 186-188).

Comments 8. Line 140, the daily dose was 2 μL which was not enough for larvae of different ages.

Response 8: Here, 20 µL of different insecticide solutions were fed to the larvae using a dropper. Modifications have been made in the text (lines 204-206).

Comments 9. Line 145-146, the final concentration of the solution was LC20/2?

Response 9: The final concentration of the solution was LC30 (lines 219-210).

Comments 10. Line 148, what the dose of the pesticide solution.

Response 10: The dose of the pesticide solution in Table 1, we have described it in the text (lines 200-203).

Comments 11. Line 152, milliliters? or mL?

Response 11: milliliters refer to mL, we have revised and rewritten this sentence in the text (lines 226-230).

Comments 12. In Materials and Methods, please supply the detail information of the Instruments and equipment, such as manufacturer and equipment model etc.

Response 12: We have provided details of the instrument and equipment in Materials and Methods.

Comments 13. Line 162, what is the number of intestinal tracts of wasps used for research ? Why not a whole wasp, but just choose the intestines.

Response 13: There were 15 wasp intestines used for research, with three replicates in each group (lines 236-240). We mentioned in the discussion: the use of intestinal tissue for RNA sequencing in Vespa magnifica is essential because the gut plays a critical role in physiological processes like digestion, immunity, and detoxification. It is also the primary site for toxin exposure and response when studying the effects of insecticides (lines 460-466).

Comments 14. Line 166-167, as this work being done by you or biological company, if it is a company, please supply the name of the company.

Response 14: This work was completed by a biological company, and we have already supply the name of the company (lines 252-253).

Comments 15. Line 177, what is the reason for choosing CYP450 gene? The full name of CYP450 should be provide.

Response 15: We have already explained the reason for choosing CYP450 gene and have provided the full name of CYP450 (lines 273-277).

Comments 16. Line 181, the“2”in“ddH2O”should be subscripted.

Response 16: The “2”in“ddH2O” have subscripted (line 288).

Comments 17. Line 183, why GAPDH was selected as the internal reference gene? However, the Real time quantitative PCR (RT-qPCR) methods are inadequate. The authors should refer to Bustin et al. 2009 (The MIQE Guidelines: Minimum Information for Publication of Quantitative Real-Time PCR Experiments. Clinical Chemistry 55:4611-622). In general, at least two or more internal reference genes must be provided to achieve accurate normalization, because the transcript levels of reference genes are not always stable and vary with the developmental stage and organization of the insect as well as the experimental conditions.

Response 17: We have explained the selection of GAPDH as the internal reference gene and included the relevant reference (lines 277-281). Additionally, we have revised the Real-Time Quantitative PCR (RT-qPCR) methods based on the MIQE Guidelines (Bustin et al., 2009, The MIQE Guidelines: Minimum Information for Publication of Quantitative Real-Time PCR Experiments, Clinical Chemistry 55:611-622) (lines 281-291). In this study, we selected GAPDH as the sole internal reference gene. Based on existing literature and our preliminary experimental results, GAPDH showed stable expression under various experimental conditions according to transcriptome data. Additionally, GAPDH has been widely validated as a reliable reference gene in insect studies, ensuring the credibility and consistency of experimental data. Therefore, although the use of multiple reference genes is generally recommended, we believe that using GAPDH alone is sufficient for normalization in this study.

Comments 18. Line 184, supply the reference of“CT method”.

Response 18: We have supply the reference of“CT method” (line 291)

Comments 19. Line 189 and 193, supply the information about SPSS and Origin.

Response 19: We have provided the information about SPSS and Origin (lines 294-298).

Comments 20. Line 195, 220, 221, Sublethal effects of insecticides

Response 20: We have changed "sublethal insecticide" to "Sublethal effects of insecticides" (line 301, 332, 333).

Comments 21. Line 203, why does thiamethoxam LC30 cause weight gain in larvae ? and also a further discussion.

Response 21: The reason why thiamethoxam LC30 causes larval body weight gain is explained and discussed in the discussion (lines 431-437).

Comments 22. Line 211-213, this sentence is difficult to read.

Response 22: We have rewritten this sentence to make it easier to understand (lines 438-459).

Comments 23. Line 220, due to the different mechanisms of action of the different insecticides, the horizontal coordinates are classified according to the pesticides in Figure 1, while the results of the different treatments are expressed in coloured bars, which makes it easier for the reader to know the differences between treatments of the same active ingredient at different concentrations.

Response 23: We have shown the results of different treatments with pesticides in Figure 1 as colored bars to facilitate readers to understand the differences between treatments with the same active ingredient at different concentrations.

Comments 24. Line 228-229, this sentence is not clear, is it the RNA of all the samples or just one of them?

Response 24: RNA here refers to the total RNA of each sample and has been modified in the text (line 340-341).

Comments 25. Line 268-272, this section is duplicated in the Materials and Methods section, and it is sufficient to state only the results, which are suggested for deletion.

Response 25: We have removed this section and only state the results (lines 393-394).

Comments 26. Line 302-304, it is possible to analyse the causes of the decrease in pupation rates in the different treatment groups.

Response 26: We have analyzed the reasons for the decrease in pupation rate in different treatment groups in the discussion (lines 413-416)

Comments 27. Line 306, the object of reference 31 and 32 is not Apis mellifera mellifera, please check.

Response 27: We have checked and corrected the object of reference 31 and 32 is Bombus terrestris and crabronid wasp in text (lines 427-430).

Comments 28. Line 310-313, it is appropriate to talk about the effects of insecticides on the three enzymes in terms of their functions.

Response 28: We have already discussed the effects of pesticides on these three enzymes functionally (lines 438-459).

Comments 29. Line 316, can ROS changes be caused by four insecticides? add the appropriate literature description.

Response 29: We have added descriptions of ROS changes caused by four pesticides to the Discussion and have added references (lines 438-459).

Comments 30. Line 325, this citation [38] is inappropriate and more literature needs to be reviewed and the effects of pharmaceuticals on their energy metabolism discussed.

Response 30: We have reviewed additional literature and eliminated inappropriate literature, and have discussed the effects of four insecticides on energy metabolism in bumblebees (lines 470-525).

Comments 31. Line 329-339, In-depth analysis of transcriptome results for each insecticide in relation to its mechanism of action.

Response 31: We have conducted an in-depth analysis of the transcriptomic results of each pesticide and its mechanism of action (lines 470-525).

Comments 32. Line 334, is there a statistics on food consumption in this study?

Response 32: This study did not include statistics on food consumption, and relevant discussion has been removed from the article.

Comments 33. Line 342, Is there any relevant literature on Vespa magnifica has fewer detoxification genes compared to other species. If so, please add.

Response 33: We have not yet found relevant literature saying that wasps have fewer detoxification genes than other species, so this content has been removed from the discussion.

Comments 34. I don't really agree with the conclusion that the alteration of these genes (CYP4, CYP4V, CYP6, CYP61, CYP9, PPIB, GST, and UTG) leads to immune disorders and impact the overall physiological state of Vespa magnific.

Response 34: Changes in these genes (CYP4, CYP4V, CYP6, CYP61, CYP9, PPIB, GST and UTG) can only indicate that Vespa magnific mitigate the effects of pesticide exposure through the CYP450 system and related detoxification mechanisms, but cannot cause immune disorders and affect Overall review status. We have made changes in the text (lines 526-535).

Comments 35. The format of the references does not meet the requirements of the journal and needs to be modified. Such as volume and issue need to be in italics, a comma between the page number and the volume and issue. Please check them one by one.

Response 35: We have changed all references to comply with journal requirements (lines 537-560).

Reviewer 3 Report

Comments and Suggestions for Authors

I appreciate the opportunity to evaluate this manuscript.

In my opinion, major revision is needed in order that the manuscript can be published. I will address these issues in a point-by-point section after the general comments written bellow.

This paper investigates the effects of sublethal pesticide exposure on the non-target insect Vespa magnifica, employing physiological and transcriptomic analyses. The study explored four pesticides - thiamethoxam, avermectin, chlorfenapyr, and β-cypermethrin - and how these impacted various biological aspects of Vespa magnifica.

General comments

First of all, the manuscript needs English editing. Consistently write the species name in italic.

Introduction

The introduction is written inconsistently. There is too much information about each pesticide individually. However, it could be improved by including a more detailed discussion of existing research on other non-target insect species affected by similar pesticides, to better contextualizing the significance of V. magnifica in this field.

Line 90-99 contains a description of the research and conclusions that do not belong in the introduction. However, the aim of the work is missing, so authors must write the aim which is an integral and important part of the introduction section.

Material and methods

Point 2.2 does not belong to the Material and methods section, but it could be written in the introduction. Points 2.5, 2.6 and 2.7 are not sufficiently explained.

Results

The figures are unclear and insufficiently explained, especially figure number 4. This should be sorted out.

Discussion

Discussion is written in a good manner.

Conclusion

The conclusion must be based on the results of this research, without discussing the data of other authors.

The conclusions should be written more concisely and concretely.

References

The references are properly cited by using more recent papers.

Comments on the Quality of English Language

Extensive editing of English language required.

Author Response

Dear reviewer,

Thank you for reviewing our manuscript and for the valuable comments, which greatly helped us to improve the manuscript. The manuscript was carefully revised and point-by-point response was listed below. We hope that your comments have been addressed accurately. The revised manuscript was marked with yellow color and the responses were presented in blue text.

Comments general comments

First of all, the manuscript needs English editing. Consistently write the species name in italic.

Response general comments: We have edited the manuscript in English.  The full text has also been checked to ensure that species names are always written in italics.

Comments introduction

The introduction is written inconsistently. There is too much information about each pesticide individually. However, it could be improved by including a more detailed discussion of existing research on other non-target insect species affected by similar pesticides, to better contextualizing the significance of V. magnificain this field.

Line 90-99 contains a description of the research and conclusions that do not belong in the introduction. However, the aim of the work is missing, so authors must write the aim which is an integral and important part of the introduction section.

Response introduction:

(1) The introduction is written inconsistently. There is too much information about each pesticide individually. However, it could be improved by including a more detailed discussion of existing research on other non-target insect species affected by similar pesticides, to better contextualizing the significance of V. magnificain this field.

We have rewritten the introduction to improve it by including a more detailed discussion of existing research on other non-target insect species affected by similar pesticides (lines 80-145).

(2) Line 90-99 contains a description of the research and conclusions that do not belong in the introduction. However, the aim of the work is missing, so authors must write the aim which is an integral and important part of the introduction section.

We have removed descriptions of the study and conclusions that do not belong in the introduction and have added the aim of the study (lines 147-158).

Comments material and methods

Point 2.2 does not belong to the Material and methods section, but it could be written in the introduction. Points 2.5, 2.6 and 2.7 are not sufficiently explained.

Response material and methods: We have put 2.2 into the introduction (lines 48-61).  And 2.5, 2.6 and 2.7 are fully explained.

Comments results

The figures are unclear and insufficiently explained, especially figure number 4. This should be sorted out.

Response results: We have reorganized the description of the image to make the content of the article clearer.

Comments discussion

Discussion is written in a good manner.

Response discussion: We sincerely appreciate the reviewer's recognition of the discussion section. Based on the feedback, we have revised and expanded the discussion to make the manuscript clearer and more comprehensive.

Conclusion

The conclusion must be based on the results of this research, without discussing the data of other authors.

The conclusions should be written more concisely and concretely.

Response conclusion: We have rewritten the discussion to make it more concise and specific (lines 537-560).

References

The references are properly cited by using more recent papers.

Response reference: We have updated some of the references to ensure they are more recent and provide stronger, more comprehensive support for our article.

Round 2

Reviewer 1 Report

Comments and Suggestions for Authors

The authors have made significant improvements to the article, making it clearer and more robust. Considering these enhancements and their responses to the reviewers' questions, I recommend this article for publication.

Reviewer 2 Report

Comments and Suggestions for Authors

The revision well addressed my previous concerns. But the conclusion section is too long and needs further simplification

Comments on the Quality of English Language

The English writing has also been greatly revised and improved